# The BE COOL Treatments (Batroxobin, oxygEn, Conditioning, and cOOLing): Emerging Adjunct Therapies for Ischemic Cerebrovascular Disease

**DOI:** 10.3390/jcm11206193

**Published:** 2022-10-20

**Authors:** Siying Song, Hao Wu, Xunming Ji, Ran Meng

**Affiliations:** 1Department of Neurology, Xuanwu Hospital, Capital Medical University, Beijing 100053, China; 2Advanced Center of Stroke, Beijing Institute for Brain Disorders, Beijing 100053, China; 3Department of China, America Institute of Neuroscience, Xuanwu Hospital, Capital Medical University, Beijing 100053, China; 4Department of Neurosurgery, Xuanwu Hospital, Capital Medical University, Beijing 100053, China

**Keywords:** remote ischemic conditioning (RIC), normobaric hyperoxia (NBO), hypothermia, batroxobin, ischemic cerebrovascular disease

## Abstract

Ischemic cerebrovascular disease (ICD), the most common neurological disease worldwide, can be classified based on the onset time (acute/chronic) and the type of cerebral blood vessel involved (artery or venous sinus). Classifications include acute ischemic stroke (AIS)/transient ischemic attack (TIA), chronic cerebral circulation insufficiency (CCCI), acute cerebral venous sinus thrombosis (CVST), and chronic cerebrospinal venous insufficiency (CCSVI). The pathogenesis of cerebral arterial ischemia may be correlated with cerebral venous ischemia through decreased cerebral perfusion. The core treatment goals for both arterial and venous ICDs include perfusion recovery, reduction of cerebral ischemic injury, and preservation of the neuronal integrity of the involved region as soon as possible; however, therapy based on the current guidelines for either acute ischemic events or chronic cerebral ischemia is not ideal because the recurrence rate of AIS or CVST is still very high. Therefore, this review discusses the neuroprotective effects of four novel potential ICD treatments with high translation rates, known as the BE COOL treatments (Batroxobin, oxygEn, Conditioning, and cOOLing), and subsequently analyzes how BE COOL treatments are used in clinical settings. The combination of batroxobin, oxygen, conditioning, and cooling may be a promising intervention for preserving ischemic tissues.

## 1. Introduction

Ischemic cerebrovascular disease (ICD) is the pathological process in which an area of the brain is temporarily or permanently affected by ischemia involving one or more cerebral blood vessels. ICD can develop from a variety of causes [1,2,3]. Despite the lack of consensus, ICD is often classified according to the onset time (acute/chronic) and the type of cerebral blood vessel involved (artery or venous sinus). Therefore, classifications include acute ischemic stroke (AIS)/transient ischemic attack (TIA) [4,5], chronic cerebral circulation insufficiency (CCCI), acute cerebral venous sinus thrombosis (CVST), and chronic cerebrospinal venous insufficiency (CCSVI) [1,2,3,6]. In clinical settings, cerebral arterial and cerebral venous ischemia are typically regarded as separate pathological processes due to different etiologies. Therefore, each has different traditional treatment methods [7]. However, the pathogenesis of cerebral arterial ischemia may be correlated to cerebral venous ischemia since both forms of ischemia can reduce cerebral perfusion [6,8]. Therefore, from a broader perspective, it is apparent that the core treatment goals for both arterial and venous ICD are cerebral perfusion recovery, ischemic injury reduction, and preservation of the neuronal integrity in the affected ischemic region as soon as possible [9].

Long-term CCCI due to large vessel stenosis, atherosclerotic stenosis, or the occlusion of the intracranial and extracranial large arteries is the initiating factor of cerebral arterial ischemia, which leads to AIS or TIA [10]. In appropriate patients, AIS treatment involves tissue plasminogen activator (tPA)/tenecteplase (TNK) fibrinolysis (within 4.5 h from onset) or thrombectomy. CCCI treatments, such as the standardized use of anti-platelet, hypertension, hyperlipidemia, and type 2 diabetes medications, mainly focus on the secondary prevention of an aggressive atherosclerosis formation [11]. However, even after such treatments, the incidence of CCCI/AIS and the recurrence rate of AIS are still very high, ranging from 5.7% to 51.3% [12,13].

Treatment of ischemia in the cerebral venous sinus or cortical veins is limited to anti-coagulation for the long-term management of chronic CVST or thrombectomy in acute CVST. CCSVI due to chronic CVST, cerebral cortical vein thrombosis (CCVT), enlarged arachnoid granules, and the compression of bones is challenging to treat due to the low anticoagulant concentration in chronic venous thrombosis or small cerebral cortical veins, as well as limited options for surgical correction [14].

Novel pharmaceutical and non-pharmaceutical methods for recovering cerebral perfusion and neuroprotection are highly sought. Although some drugs were promising in preclinical models for improving cerebral perfusion and rescuing ischemic areas, their use has not translated to clinical settings due to the failure to include stroke models with significant comorbidities and a lack of testing in older animal models [9,15,16]. Still, several clinical trials have yielded positive results and indicated beneficial effects in stroke patients, such as the use of human urinary kallidinogenase (HUK) (NCT03431909 Phase IV China), edaravone (NCT02430350 Phase III China/Japan), and nerinetide (NA-1) (NCT02930018 Phase III Canada and USA) [17]. Most intriguingly, in multiple clinical trials, batroxobin has recently drawn attention due to its protective effects in the cerebral artery and venous ischemia [18,19,20]. Non-pharmaceutical neuroprotective methods, including normobaric oxygen therapy (NBO), remote ischemic conditioning (RIC), and hypothermia, have been tested in several clinical trials, and the beneficial effects remain controversial for stroke prevention and recurrence [21,22,23].

This review aims to provide a comprehensive assessment of common effective therapies for cerebral arterial and venous ischemia. Four novel pharmaceutical and non-pharmaceutical treatments are grouped into one combined therapy, referred to as the **BE COOL** (**Batroxobin, oxygEn, Conditioning, and cOOLing**) treatments.

## 2. Batroxobin

Batroxobin, isolated from *Bothrops atrox moojeni* snake venom, is widely used in treating AIS and CVST due to its role in promoting thrombolysis, recurrence of thrombus, and neuroprotection. Batroxobin could be an effective adjunctive therapy to traditional anti-coagulation and anti-platelet treatments due to its proven safety and low incidence of hemorrhage transformation in multiple preliminary clinical studies [14,19,20,24,25]. Previously, our team systematically summarized the clinical effects and related mechanisms of batroxobin in various vascular diseases [18] Therefore, this review focuses on the protective effects and mechanisms of batroxobin in ICD.

### 2.1. Possible Neuroprotective Mechanisms of Batroxobin

In animal models of cerebral ischemia or ischemia-reperfusion, batroxobin was involved in the inhibition of neuron apoptosis, reduction of cerebral edema, a decrease in hemorrhagic transformation, and the recovery of cerebral perfusion to the infarcted sites [18]. Several pathophysiological mechanisms based on preclinical studies were proposed regarding the neuroprotective effects of batroxobin. First, batroxobin directly targets fibrinogen, a significant component of clots; therefore, batroxobin could decrease the deposition of fibrinogen to form fibrin [26,27,28,29]. Another neuroprotective effect of batroxobin was the direct up-regulation of myelin basic protein (MBP), which is vital in nerve myelination in the nervous system [27]. Moreover, batroxobin activates endothelial cells to release endogenous tPA, promoting thrombolysis. Batroxobin could also increase the bioactivity of superoxide dismutase (SOD) and eliminate oxygen-free damage to the infarcted area [18]. Lastly, batroxobin may inhibit the expression of various pro-inflammatory markers in peripheral serum and the injured cerebral region (e.g., serum tumor necrosis factor-alpha (TNF-α), heat shock proteins 32 and 70 in the cerebral ischemic region, and complements C3d and C9 in the cerebral perihematomal area) [30,31,32].

Batroxobin, as an adjunctive therapy was more widely tested in patients with AIS [24,33,34,35,36,37,38,39], and more recently, a few clinical studies extended the application of batroxobin in CCCI [25] and CVST [19,20]. Most clinical studies have shown positive effects in prognosis after batroxobin usage; however, these studies are limited to small sample sizes, population selection biases (most studies are based in Asian countries), unblinding and unrandomized procedures, and various treatment regimens between different studies. Therefore, interpretations of the findings should be made with caution.

### 2.2. Batroxobin and AIS

Several clinical studies evaluated the efficacy and safety of different treatment regimens of batroxobin in AIS patients [24,33,34,35,36,37,38,39]. The use of batroxobin alone decreased fibrinogen concentrations and erythrocyte aggregability, reduced stroke recurrence rates, and produced more significant improvements in neurological assessments [18]. Interestingly, AIS/TIA patients with hyperfibrinogenemia saw a more substantial benefit when using batroxobin than that with normal serum fibrinogen levels, perhaps due to the elimination of excessive fibrinogen [38]. Moreover, the combined use of batroxobin and the standard post-stroke treatment (aspirin and statins) proved safe in several clinical studies and improved cerebral hemodynamics [33,39,40].

The combined use of edaravone and batroxobin was also evaluated in several small-size randomized case–control studies in the Chinese population, including two studies that enrolled patients with progressive AIS [24,35,37]. Although the exact mechanism of action in the edaravone treatment of AIS is unknown, its therapeutic effect may be due to its known antioxidant properties [17]. Post-stroke oxidative stress is a part of the process that damages the infarcted area. Wu et al. observed decreased neurological deficit scores and serum fibrinogen levels in AIS patients treated with either batroxobin alone or in combination with edaravone, with no adverse effects in either group. Further, the combination group had a higher effective rate than the group that received only batroxobin [24]. Ren et al. and Wang et al. included patients with progressive AIS [35,37]. Similar results with Wu et al. were also founded. However, these results should be interpreted with caution due to the limited sample size and variable treatment duration (30 mg intravenous edaravone for 10 days or 30 mg intravenous edaravone for 14 days). Evaluating the safety and efficacy of batroxobin in more extensive randomized clinical trials is necessary.

### 2.3. Batroxobin and CCCI

Unlike AIS, CCCI exhibits a chronic degenerative course in the absence of acute symptoms, leading to a delayed diagnosis in many patients [10]. Without treatment or intervention, CCCI can lead to adverse events such as cognitive impairment, depression, and AIS. Due to a lack of effective treatments or drugs, CCCI treatment is limited to diet and physical activity modification and pharmaceutically controlling the risk factors of hyperlipidemia, high blood pressure, or diabetes mellitus [3]. Zhai et al. evaluated the effects of batroxobin in patients with vascular cognitive dysfunction. Compared to the control group (standardized use of aspirin only), patients who received a combination of aspirin and batroxobin displayed a considerable improvement in cognitive function and quality of daily life [25].

### 2.4. Batroxobin and Acute CVST

Long-term standardized anti-coagulation therapy is standard for both acute CVST and CCSVI. The pathogenesis of cerebral venous ischemia differs from that of cerebral arterial ischemia [1,7]. Risk factors contributing to acute or chronic cerebral venous ischemia are related to the Virchow triad, including vascular damage, stasis, and hypercoagulation. As a defibrinogenating agent, batroxobin has been combined with anticoagulatory drugs in two acute CVST studies [19,20]. Ding et al. evaluated acute CVST recanalization after batroxobin and anti-coagulation (subcutaneous injection of low molecular weight heparin (LMWH, 0.4 mg/q12 h) bridged with warfarin (3 mg/d for 5–7 days), followed by oral warfarin (3 mg/d), to maintain the international normalized ratio (INR) between 2 and 3 using both magnetic resonance venography (MRV) [20] and high-resolution magnetic resonance imaging (HR-MRI) [19]. The authors demonstrated that batroxobin is a safe and effective adjunct agent able to promote CVST recanalization. Furthermore, Ding et al. divided patients into subgroups based on fibrinogen levels [20]. Acute CVST patients with higher fibrinogen levels showed greater recanalization from batroxobin. However, these two small-size prospective studies by Ding et al. enrolled acute CVST patients in the same medical center, increasing the possibility of selection bias, and observer bias could be present due to a lack of blinding and randomization.

### 2.5. Batroxobin and CCSVI

Zamboni and colleagues first defined CCSVI as a chronic state of impaired cerebral or cervical venous drainage [41]. Because the symptoms of CCSVI (e.g., headache, tinnitus, and head noises (whooshing noises inside the head)) are non-specific, missed diagnosis and misdiagnosis rates are relatively higher than other types of ICD. CCSVI has been mistakenly confirmed in apparently healthy people after a neurological screening using MRV or computed tomographic venography (CTV) [3]. CCSVI may induce venous refluxes and cerebral venous hypertension, resulting in the disruption of brain-blood barrier (BBB) integrity, peri-venous iron accumulation, and decreased cerebral brain flow (CBF), leading to chronic cerebral hypoxia, inflammatory cell infiltration of the brain parenchyma, and even localized inflammation [1,42].

CCSVI caused by chronic CVST or multiple CCVT is challenging to treat due to insufficient concentrations of oral anti-coagulation around the clots. Song et al. enrolled a small group of CCSVI patients with chronic cerebral venous thrombosis (CVT) and found that the combined use of batroxobin and novel oral anticoagulants (NOAC) had decreased time to symptom relief, higher recanalization rates of chronic CVT, and less recurrence of acute CVT [14]. However, this case series was a retrospective study with a small sample size, and the oral anticoagulant regimens varied between patients. Therefore, future clinical studies are needed to evaluate the effects of batroxobin on CCVSI further.

## 3. Normobaric Oxygen

Normobaric oxygen (NBO), given via a facial mask or nasal cannula, has been safety and efficacy tested in protecting ischemic brain tissue with proven protective effects. However, the high heterogeny of NBO intervening modes (initiation time, fraction of inspired oxygen (FiO_2_), flow velocity, device (facial mask or nasal tube), and duration) led to different efficacy results and debate over its clinical implementation [43,44]. Based on the former meta-analyses and systematic reviews of NBO in human stroke studies and the use of NBO in clinical practice [43], this section discusses NBO’s application in rescuing both acute and chronic ischemic damage to the brain tissue.

### 3.1. Possible Neuroprotective Mechanisms of NBO 

Oxygen supplementation is a common adjuvant therapy for various diseases. Based on the inhaled oxygen pressure (greater than or equal to atmospheric pressure, 1 ATM = 101.325 kPa), oxygen therapy can be classified into NBO and hyperbaric oxygen (HBO) [45]. HBO is not widely used due to the expensive cost of implementation and its potentially damaging side effects on lung tissues. However, the neuroprotective role of NBO (given via facial mask or nasal cannula) has been suggested in treating strokes, as it is non-invasive and easy to administer in prehospital settings. Despite there being no significant increase of the total arterial oxygen content, NBO safely and effectively increased the ischemic penumbral partial pressure of oxygen (PtO_2_) in various rodent models of ischemic stroke [46]. Moreover, NBO treatment did not cause oxidative stress, which is common in theory after reperfusion [47].

### 3.2. NBO and AIS

Our team previously performed a meta-analysis to evaluate the effect of NBO in AIS patients based on eleven prospective RCT studies and found that existing data trended toward treatment-related benefits, which was encouraging for researchers in this field [43]. The promising value of NBO needed further examination based on standard modes of NBO intervention. Most studies concluded that NBO intervention initiated within 12 h of stroke onset resulted in a better prognosis. Oxygen flow velocity in these studies was usually set as 10 L/min and maintained for more than 12 h. A systematic review by Mahood et al., which included fifteen articles on NBO therapy for AIS patients, comprehensively reviewed the outcomes of mortality, symptom relief, neurological function, and neuroimaging improvements in AIS patients after NBO intervention [48]. Although the difference between the NBO group and the control group was not considered significant due to the small sample size and heterogeneity of NBO intervention modes, the authors observed a neuroprotective trend associated with NBO treatment. No oxygen-related adverse events were reported, and the incidence of serious adverse events was lower in the NBO group.

### 3.3. NBO and CCCI

Considering the shared underlying mechanisms between the penumbra during a stroke and the ischemic–hypoxic brain tissues in CCCI, we speculate that NBO may be a promising therapeutic strategy for attenuating short-term symptoms or improving long-term clinical outcomes among CCCI patients [44]. However, few studies have evaluated the efficacy of NBO in CCCI patients. Our team was the first to test the neuroprotective effects of NBO in CCCI. Additionally, Ding et al. innovatively utilized EEG recordings to detect early changes in CCCI-related EEG anomalies more precisely after NBO [49]. Further, NBO ameliorates CCCI-related EEG anomalies, including attenuating abnormal high-power oscillations and the slow paroxysmal activities associated with CCCI. A more precise method of detection, such as EEG, may be helpful in the evaluation of NBO efficacy. Experimental and clinical studies are necessary to shed more light on the application of NBO in CCCI.

## 4. Remote Ischemic Conditioning (RIC)

RIC is the application of reversible episodes of ischemia and reperfusion in one vascular bed, tissue, or organ, conferring global protection and rendering remote tissues and organs resistant to ischemia/reperfusion injury. Upper limb RIC has mainly been applied in clinical trials. In its first description by Murry et al. in 1986, RIC was viewed as a promising cardioprotective technique [50]. The CONDI-1 trial (*n* = 333) [51], RIC-STEMI trial (*n* = 258) [52], and LIPSIA CONDITIONING study (*n* = 696) [53] provided evidence that RIC could reduce circulating biomarkers of myocardial necrosis and edema and could improve cardiac function. Recently, the role of RIC in improving STEMI patient prognosis has been questioned mainly due to the nonbeneficial effects of RIC illustrated by Derek et al. in the CONDI-2/ERIC-PPCI trial, a large and appropriately powered randomized controlled trial (*n* = 5401) [22,54,55]. However, RIC has been gradually used for post-stroke rehabilitation and prevention of recurrence due to its proven beneficial effect in randomized clinical trials (though with a small sample size of fewer than 100 patients) and user-friendly design (easy to carry on, non-invasive, and cost-effective) [56,57]. Several previous reviews have demonstrated the protective mechanisms in ICD based on the interaction of neural, humoral, and immunological systems induced by RIC [10,58]. This review briefly summarizes the primary mechanism of RIC based on previous reviews and original studies and introduces the clinical application of RIC in ICD.

### 4.1. Possible Neuroprotective Mechanisms of RIC 

Based on data obtained during animal studies, RIC protection consists of two stages [59,60]. The first (acute) response starts from 2 to 4 h post-RIC, which primarily presents an immediate increase of blood flow in cutaneous microcirculation via the enhancement of nitric oxides (NO)-synthesis. Then, the second protection window occurs from 12 to 24 h after RIC [61]. Protective measures within this phase can be attributed to three closely interacting major signaling pathways, including the humoral pathway mediated by NO, erythropoietin (EPO), adenosine, and heme oxygenase-1 (HO-1) [62,63]; the immunological pathway mediated by the downregulation of pro-inflammatory markers (e.g., IL-6 and IL-1) and the suppression of inflammation by elevating peripheral monocytes/lymphocytes and inhibiting neutrophil activation; and the neurological pathway mediated by direct stimulation of peripheral sensory nerves [59]. 

### 4.2. RIC and AIS

Despite the proven safety and feasibility of RIC in AIS in several clinical trials, data regarding the protective effect of RIC in patients with AIS in prehospital or emergency settings are still inconclusive. Hougaard et al. found that paramedic-administered RIC given before the intravenous administration of tPA reduced the rate of severe stroke and risk of tissue infarction. At the same time, neuroimaging indicators of penumbral salvage, final infarct size, and infarct growth did not improve after RIC use [64]. Additionally, Che et al. indicated that RIC administration after intravenous tPA did not improve neurological function recovery [65]. For AIS patients without tPA administration, England et al. demonstrated that using RIC may improve neurological outcomes, and protective mechanisms may be mediated through HSP27 [66].

### 4.3. RIC and CCCI

Long-term RIC intervention could provide protection for CCCI patients. Meng et al. firstly demonstrated that bilateral RIC administered twice daily for over 300 consecutive days might improve cerebral perfusion and reduce recurrent strokes in patients with symptomatic atherosclerotic intracranial arterial stenosis (IAS) [67]. The authors further evaluated the safety and efficacy of RIC in patients over 80 years of age with symptomatic atherosclerotic IAS, as elderly patients have a high incidence of adverse events with anti-platelet administration and are rarely encouraged to undergo endovascular stenting for symptom relief [68]. Anti-inflammation and anti-coagulation effects were observed after 6 months of RIC. Furthermore, RIC reduced the high-sensitivity C-reactive protein (hs-CRP), interleukin-6 (IL-6), plasminogen activator inhibitor-1, leukocyte count, platelet aggregation, and elevated the plasma tissue plasminogen activator. Stroke occurrence and recurrence also decreased in elderly patients with symptomatic IAS after long-term RIC intervention. 

Moreover, Meng et al. found that long-term RIC could also reduce the size of white matter lesions and improve cognitive function in the same cohort of elderly patients [69]. Patients with severe carotid artery stenosis could also benefit from RIC before carotid artery stenting. Ischemic brain injury secondary to CAS was highly reduced when RIC was administered twice daily for two weeks before stenting. However, the aforementioned preliminary studies were based on patients from the Chinese population enrolled in the same medical center. Therefore, the beneficial effects of long-term RIC in CCCI should be interpreted with caution.

### 4.4. RIC and CCSVI

Anti-coagulation and symptomatic relief (e.g., headache, tinnitus, and depression/anxiety) remain the primary treatment option for CCSVI patients, yet the overall symptom relief rate is low, negatively impacting patients’ quality of life [1]. Studies on the safety and efficacy of RIC in CCSVI are still scarce. Based on previous findings of long-term RIC-induced anti-inflammation and anti-coagulation effects in IAS, Song et al. tested whether these effects were achievable in CCSVI patients. The authors enrolled a group of CCSVI patients (*n* = 248), including chronic CVST-related CCSVI and chronic internal jugular venous thrombosis (IJVS)-related CCSVI [70]. However, there was no significant change in IL-6 or hs-CRP inflammatory markers, nor in coagulatory markers after a single administration of RIC. Studies with larger sample sizes and extended RIC intervention periods are needed in the future.

## 5. Hypothermia 

The average core body temperature is approximately 37 °C in humans; hypothermia is defined as a body core temperature below 35 °C. Therapeutic hypothermia is thought to be one of the most robust neuroprotective strategies available [71]. There are two major cooling/hypothermia interventions: endovascular hypothermia and surface cooling. Although the application of hypothermia in clinical settings can be traced back to 3500 BC (first recommended in the Edwin Smith Papyrus, an ancient Egyptian treatise on medicine and surgery) [72], hypothermia has only been accepted as a promising treatment in modern medicine since the late 1980s. Busto and colleagues reported marked protection induced by lowering the brain temperature by only a few degrees [73]. After years of preclinical and clinical studies, therapeutic hypothermia for stroke has been shown to be feasible but has yet to be definitively proven effective [74].

### 5.1. Possible Neuroprotective Mechanisms of Hypothermia

The neuroprotective mechanisms of hypothermia in AIS have been thoroughly studied over the past few decades [75]. Four significant protective effects were indicated in previous reviews based on experimental studies. First, hypothermia reduces cerebral metabolic rates by lowering oxygen consumption, glucose utilization, and lactate levels [76,77,78]. During the systematic cooling process, intracellular ATP levels are preserved by maintaining an ion gradient, thereby avoiding calcium overload and subsequent cell damage [79]. Furthermore, apoptosis is primarily affected by hypothermia, during which both mitochondrial-based intrinsic pathways and receptor-mediated extrinsic pathways are down-regulated [80,81,82]. Moreover, the post-stroke inflammatory storm is one of the primary reasons for damage to ischemic brain tissues [83,84,85]. The anti-inflammatory effects of cooling were mediated by suppressed astrocyte and microglial activation and leukocyte infiltration advantageously reduced inflammatory mediators, adhesion molecules, and pro-inflammatory cytokines [86,87,88]. Intriguingly, hypothermia also protected the structural integrity of the blood–brain barrier (BBB) and white matter, which perform essential roles in brain homeostasis.

### 5.2. Hypothermia and AIS

Therapeutic hypothermia in AIS patients has been widely tested in clinical trials using endovascular hypothermia (*n* = 18) and surface cooling (*n* = 22) [75]. The safety and feasibility of using hypothermia and tPA were verified in several clinical studies. However, the beneficial effects of therapeutic hypothermia were still inconclusive due to differences in the stroke severity and recanalization rate of enrolled patients and the use of various critical parameters during hypothermic therapy (e.g., starting time, duration, depth, rewarming speed, and cooling methods). Several studies demonstrated that surface hypothermia could help control elevated intracranial pressure and decrease glutamate, glycerol, lactate, and pyruvate in the “tissue at risk” area of the infarct, improving neurological function [89,90]. Nevertheless, the invasiveness of endovascular hypothermia can trigger more side effects, such as shivering, hyperglycemia, decreased cardiovascular function, increased risk of post-stroke infection, and inhibition of coagulation factors [71]. In a systematic review, Wu et al. concluded that a fast starting time, mild depth, and short duration in therapeutic hypothermia could provide long-term neuroprotective effects. Additionally, a slow 12-h rewarming may be appropriate in clinical practice [75]. Conclusive efficacy trials assessing therapeutic hypothermia combined with reperfusion therapies in AIS are ongoing [91].

### 5.3. Beyond Traditional Hypothermic Therapies

Traditional hypothermic therapies primarily refer to endovascular cooling and surface cooling. Some clinical studies indicated the unsatisfactory efficacy of these cooling methods. In recent years, several novel hypothermic therapies have emerged, including pharmacologic cooling, selective brain cooling, intra-arterial selective cooling infusion, epidural cooling, subdural cooling, subarachnoid cooling, and retrograde jugular venous cooling. A detailed description of all these new cooling methods was summarized in our previous review [75]. No matter how elaborate novel cooling terms may appear, the principal goal of improving the cooling process has not changed: to achieve more rapid and profound local brain hypothermia.

## 6. BE COOL (Batroxobin, oxygEn, Conditioning, cOOLing) Treatments to Preserve the Ischemic Brain Tissue

This review summarized the neuroprotective effects of batroxobin, NBO, RIC, and hypothermia in ICD from cerebral artery and cerebral venous ischemia, as well as their combined use with tPA on AIS. Thus, the combination of batroxobin, NBO, RIC, and hypothermia may be an ideal early chronic cerebral ischemia or stroke treatment to preserve ischemic tissue. Several preclinical studies evaluated the therapeutic effects of combining NBO with mild hypothermia (MH) in thromboembolic strokes in a rat model [92,93,94,95]. Ji et al. firstly demonstrated that the therapeutic effect of tPA was enhanced when used in combination with NBO and MH [92]. Then, Cai and colleagues found that post-stroke brain damage could be prevented via the modulation of protein kinase C-protein kinase B-nitric oxide metabolite (PKC-AKT-NOX), hyperglycolysis reduction, and pyruvate dehydrogenase complex modulation [93,94,95]. Interestingly, a preliminary clinical study by Hao et al. found that the combined use of batroxobin and MH may reduce neurological deficits at day 14 and day 21 in patients with AIS (*n* = 45) [34]. However, this study was somewhat limited in sample size and has a lack of blinding and randomization. Moreover, not enough data identified the neuroprotective effects of batroxobin, NBO, RIC, and hypothermia in AIS with TNK. The degree of the clinical promise of the BE COOL treatments associated with their effectiveness, safety, low cost, and ease of administration necessitates extensive randomized controlled studies to investigate any combined neuroprotective effects further.

## Data Availability

Not applicable.

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
