# Peer review of "The BE COOL Treatments (Batroxobin, oxygEn, Conditioning, and cOOLing): Emerging Adjunct Therapies for Ischemic Cerebrovascular Disease"

_jcm, 2022, doi:10.3390/jcm11206193_

Round 1
Reviewer 1 Report
GENERAL COMMENTS
This is an interesting narrative review of cutting-edge research on an under-researched topic, i.e. emerging therapies for brain ischemia. The authors take on the difficult task of reviewing the evidence on multiple original articles focusing on batroxobin, oxygen, conditioning, and hypothermia as potential therapeutical intervention, mainly as add-ons to existing therapies.
While it is not easy to discuss such an exploratory topic, some points need to be addressed before the article is suitable for publication. In particular, a more unbiased and critical view is crucial. This is detailed below, but, in general, limitations of the included studies (adding details on statistical and methodological characteristics, risk of publication bias, animal studies…), potential side-effects of treatments, the study methodology, and the discussion of results need to be further details. More careful choice of the references is also needed, giving priority to studies on humans when possible, or specifying that references are on experimental models when it is the case.
ABSTRACT
1) The abstract is overall well-structured and compelling, thank you. However, as discussed below in the minor comments, a review of some syntactical errors is necessary.
2) Furthermore, neuroprotection is a different entity from reperfusion. The abstract reads “The core goal for ICD, both the arterial side and the venous side, is to recover the cerebral perfusion in the ischemic region as soon as possible. However, the current standardized therapy for either acute ischemic events or chronic cerebral ischemia was not ideal. Therefore, in this review, we discuss the neuroprotective effects” (Lines 20-22). These lines first mention recovery of cerebral perfusion, then neuroprotective effects. It should be discussed the importance of neuroprotection, besides simple reperfusion.
TITLE
The title reads “The BE COOL treatments (Batroxobin, oxygEn, Conditioning, cOOLing) emerge as adjunct therapies to ischemic cerebrovascular disease: recovery of blood perfusion from cerebral artery to cererbal vein sinus.”.
1) I think there is a typo in “cererbal“?
2) BE COOL therapies have not yet clearly emerged as adjunct therapies for brain ischemia beyond doubt, but they are emerging in recent years. I suggest modifying the title to: “The BE COOL treatments (Batroxobin, oxygEn, Conditioning, cOOLing): emerging adjunct therapies for ischemic cerebrovascular disease”
3) I suggest removing this sentence: “recovery of blood perfusion from cerebral artery to cererbal vein sinus”. It does not add important information, and these therapies are not only aimed at recovering blood perfusion.
4) I suggest changing “to” into “for” before “ischemic”.
INTRODUCTION
1) Lines 35-37 Solid references should be provided to support the classification of cerebral ischemia, since, as the authors correctly state consensus is lacking: “ICD is often classified according to the onset time (acute/chronic) and cerebral blood vessel types (artery or venous sinus), therefore, including acute ischemic stroke (AIS)/transient ischemic attack (TIA), chronic cerebral circulation insufficiency (CCCI), acute cerebral venous sinus thrombosis (CVST) and chronic cer- 38 ebrospinal venous insufficiency (CCSVI). “.
2) Line 52: “Pharmacological studies have been disappointing.“. The authors should be more precise and explain briefly why these therapies are disappointing. Similarly, they should moderate and better detail the statement “still a lack of effective treatment and drugs”. Indeed, there are (partially) effective therapies for acute ischemic stroke. The sentence at lines 52-54 attempts to better explain this, but references are lacking and “standard therapy” is too general: which standard therapy?
3) Line 67: I think there is a typo in “Baxtrobin”. Please correct also other such typos in the whole manuscript.
4) In the Introduction, as well as in the next sections (see below), it is crucial to give a more balanced viewpoint on the state of the art on neuroprotective strategies and their origins. In detail, other existing reviews on these topics (including some mention of the fact that RIC has initially been applied in cardiology with somehow disappointing results, e.g. doi: 10.3389/fneur.2021.716316), besides those published by the authors, should be mentioned and briefly discussed, specifying better what the present review adds compared to previous ones (in addition to what stated at lines 62-64):
- The neuroprotection effect of oxygen therapy: A systematic review and meta-analysis. (doi: 10.4103/njcp.njcp_315_16. PMID: 29607850).
- Remote Ischemic Conditioning in Ischemic Stroke and Myocardial Infarction: Similarities and Differences (doi: 10.3389/fneur.2021.716316. PMID: 34764925; PMCID: PMC8576053).
- Oxidative stress and pathophysiology of ischemic stroke: novel therapeutic opportunities. ( doi: 10.2174/1871527311312050015. PMID: 23469845).
MAIN BODY
1) In the Sections “Possible mechanisms of neuroprotection by Batroxobin” and “Batroxobin” there are just two references: 13 (a review from one of the co-authors) and 15. The specific original articles proving the evidence that is discussed should be cited, not a review. The existing reviews should instead be discussed in the Section “Introduction”. However, it is crucial to give a more balanced viewpoint on the state of the art as discussed above.
2) Lines 90-91: Please rephrase and review the sentence “Lastly, inhibition of pro-inflammatory markers expression (e.g. Tumor necrosis factor-alpha (TNF-α), heat shock protein and , C3d and C9)”, detailing also if these markers are expressed at CNS level or peripherally, for instance.
3) Line 94: “A total of nine clinical studies evaluated the efficacy and safety of Batroxobin in AIS patients. 21-28 “. This is a narrative review, not a systematic review. However, if authors wish to detail all existing literature on a certain point, they should also provide the reader with crucial information to interpret these findings: when where these 9 studies extracted, from which databases, using which keywords.. In general, some aspects of methodology should be provided in the Introduction.
4) In general, a review has to be unbiased and impartial. It should clearly state and discuss the limitations of the existing evidence, not only the strengths or positive findings. Thus, the authors need to provide further details on the limitations of the study they discuss, including, but not limited to, samples sizes, blinding procedures, potential conflict of interest, statistical power, risk of side effects. Importantly, results obtained on animal models should be clearly defined as experimental, and preliminary results (on small samples size, or without randomization, or without blinding) should be labelled as such across the whole papers. Below a couple of examples but most of the sections need such corrections.
For instance, at lines 104-105 the manuscript reads: “The safety of the combination of Edaravone and Batroxobin was also proved in several case-control studies.”. A more critical discussion of the limitations and strengths of these case-control studies is necessary and a more moderate viewpoint on the safety of the combination of Edaravone and Batroxobin is necessary, since its safety is far from being definitely proved. Judging from the abstract the total sample size and the methodology of these studies are major caveats. I am quite concerned by the fact that I have not found any of the three studies referenced: could you provide the DOI for references 23, 25 and 26? Is there a typo in the title of reference 23?
Or, as another example, reference 30 needs to be better discussed since it includes 40 patients only, and its findings should clearly be labeled as exploratory.
5) Line 122: there is a typo in “he risk factors ”
6) References 31 and 32 are from the same research group, and include in total less than 100 patients (assuming there is no overlap between the two studies which look quite similar). It is thus inappropriate to state that “Batroxobin, serving as a strong defibrinogenating agent, has been widely combined with anticoagulation in several acute CVST studies” since this is not a “wide” utilization of the therapy. Additionally, once again, critical discussion of limit and downsides of this evidence is needed.
7) Which is the reference for “CCSVI has been accidentally confirmed in apparently "healthy people" after a neurological screening of MRV or computed 140 tomographic venography (CTV). ” (lines 139-140)
8) Thank you for the section on HBO and NBO, it is useful and more critical. Authors may use it as a model for the other sections.
9) As above, also most of the sections on RIC needs a more balanced discussion. Authors may considering discussing this recent open-access article on the Lancet: “The broken promise of remote ischaemic conditioning” (DOI:https://doi.org/10.1016/S0140-6736(19)32047-1). The section “RIC and CCSVI” is appropriate, but please include sample sizes of the main references discussed, thank you.
10) The section is very interesting. However it is a pity that the following sentences are supported by animal studies (ref 65-67) only: “The anti-inflammatory effects of cooling were mediated by suppressing astrocyte and micro-glial activation as well as leukocyte infiltration, accompanied by reduced levels of inflammatory mediators, adhesion molecules, and pro-inflammatory cytokines.” (lines 284-287). This should be mentioned, and evidence in humans on the topic should be added, if it exists. Indeed the sentence “Moreover, the post-stroke inflammatory storm is one of the primary reasons for damage to brain ischemic tissues.” Is not supported by any reference, authors should cite recent studies on the topic, including (but not necessarily limited to):
- Thrombo-Inflammation and Immunological Response in Ischemic Stroke: Focusing on Platelet-Tregs Interaction (DOI: 10.3389/fncel.2022.955385)
- Platelet, Plasma, Urinary Tryptophan-Serotonin-Kynurenine Axis Markers in Hyperacute Brain Ischemia Patients: A Prospective Study (DOI: https://doi.org/10.3389/fneur.2021.782317)
- Inflammatory Responses After Ischemic Stroke (DOI: 10.1007/s00281-022-00943-7)
11) The sentence “Ji et al firstly demonstrated that the therapeutic effect of tPA in ischemic stroke was enhanced in combination with NBO and hypothermia” (lines 326-327) is misleading since reference 70 is a study on rats and this should be specified, also in several other cases like this in the manuscript. Please specify each time that a result is preliminary (indicatively less than 100 subjects), or on animals.
MINOR COMMENTS
The Manuscript needs a review of the English language, since some sentences are not clear or wrong.
- For instance at Line 25-26, page 1 the abstract reads “We advocate that the combination of Batroxobin, oxygen, conditioning, or cooling may be promising treatments to pre- serve the ischemic tissues.”: the subject is singular (i.e. “combination”) so the word treatments should be singular or it should be substituted by “intervention”, since the author imply that is the combination of the treatments that would be promising, and not each of them separately.
- at Line 21-22, page 1 the abstract reads “However, the current standardized therapy for either acute ischemic events or chronic cerebral ischemia was not ideal.”. This sentence is not clear: what do the authors mean by “standardized”? The therapy is not standardized. Perhaps they mean “according to guidelines”? “was not ideal” implies that now it is ideal: I think that the verb should be at the present.
- At line 39 and at following lines “we tended” is informal and inaccurate, other forms should be considered also in the rest of the manuscript. The “we” is not appropriate for an objective review, and the article is not meant to be an opinion or viewpoint article.
- At line 43 review the sentence “as both could reduce cerebral perfusion.”
- Line 66 “We put” is not the right verb in this context. Perhaps “we group/call/identify”?
- Other issues with English should be reviewed by a professional or a mother tongue.
The manuscript needs copy-editing as there are numerous typos or formal errors (e.g. “Anti-platelet” line 51 is capital and plural but “statins” is not).
Often, references are placed after a full stop (e.g. ref 13 at line 81). I suggest placing references before the full stops.
Better sub-section organization is needed, perhaps using numbers. For instance, the section “Possible mechanisms of neuroprotection by Batroxobin” and the following ones about Batroxobin should be sub-sections of the section “Batroxobin”.
What does the “*” mean in the reference 20? “Neuroprotective role of batroxobin in cardiopulmonary resuscitation rabbits*. ” (line 377)
Reviewer 2 Report
Excellent topic but English is unreadable because the verb tenses are often incorrect or fluctuating between past and present. Need a fluent native speaker to edit this paper. Also, each subtopic requires greater definition: for example, NBO, RIC, etc., for those readers who do not know or have not heard of their properties, how they work, and trials leading up to their use in animals and humans. This paper will be great after significant revision in style and grammar because the topic is of interest to many.
Author Response
Thanks so much for your valuable comments. We have carefully revised our manuscript based on your kind suggestion. All your comments are vital for the improvement of our manuscript. We have deeply revised the four major section (Batroxobin, oxygEn, Conditioning, and cOOLing).
And we have also involved a native English speaker to recheck language of the whole manuscript. We really hope that the language level has been substantially improved.
Round 2
Reviewer 2 Report
Decent updates and edits. A few lingering questions/suggestions:
1. in the abstract, WHY is therapy based on current guidelines not "ideal" exactly?
2. cerebral arterial and cerebral venous ischemia are typically regarded as separate pathological processes due to different ETIOLOGIES, not just different risk factors
3. Delete "Traditional" and just start the sentence with: "In the appropriate patient," AIS treatment involves tissue plasminogen activator (tPA) fibrinolysis (within 4.5 hours from onset), or ***TNK***, or thrombectomy...
4. change "recently drawn the most attention" to "recently drawn attention"
5. change "This review aims to provide the first comprehensive assessment" to "This review aims to provide a comprehensive assessment"
6. change "the significant component of clots;" to "a significant component of clots;"
7. change "inconsistent treatment duration" to "variable treatment duration"
8. delete the word fluctuating when talking about INR
9. head noises? Please explain.
10. hypothermia in clinical settings can be traced back to 3500 BC? By whom exactly and in what manner?
11. Might want to mention that you were unable to identify neuroprotective effects of batroxobin, NBO, RIC, and hypothermia in ICD with TNK, only with tPA on AIS (perhaps not enough data)
